# Experience of living with chronic pain in conjunction with surgery for ulnar nerve entrapment at the elbow—A qualitative study

**Alice Giöstad** [1]*, **Ingela K. Carlsson**[2,3], **Lars B. Dahlin**[1,2,3], **Erika Nyman**[1,4]

**1** Department of Biomedical and Clinical Sciences, Linköping University, Linköping, Sweden, **2** Department of Hand Surgery, Skåne University Hospital, Malmö, Sweden, **3** Department of Translational Medicine – Hand Surgery, Lund University, Malmö, Sweden, **4** Department of Hand Surgery, Plastic Surgery and Burns, Linköping University Hospital, Linköping, Sweden

* alice.giostad@regionstockholm.se

**Data Availability Statement:** Public access to research data on humans is restricted by the Swedish Authorities (Public Access to Information and Secrecy Act; https://www.government.se/

## Abstract

### Purpose

Pain in conjunction with surgery for ulnar nerve entrapment at the elbow is seldom highlighted in the literature. This study aimed to explore patients' experiences of living with chronic pain (≥3 months duration) in conjunction with surgery for ulnar nerve entrapment at the elbow, the consequences and the coping strategies applied.

### Material and methods

In-depth interviews were conducted with 10 participants aged 18–60 years. The narratives were analyzed using an inductive approach and content-analysis.

### Results

The analysis revealed seven main categories: "*Physical symptoms/impairments*" and "*Mood and emotions*" comprise symptoms caused by ulnar nerve entrapment at the elbow and chronic pain; "*Consequences in daily life*" includes challenges and obstacles in every-day life, impact on leisure activities and social life; "*Struggling with self-image*" embraces experiences closely related to identity; "*Coping strategies*" covers adaptive resources; "*Experience of relief* "describes perceived improvements; "*Key message for future care*" comprises important aspects for healthcare providers to consider.

### Conclusions

The results clarify the need for healthcare personnel to adopt a biopsychosocial approach when treating patients with ulnar nerve entrapment at the elbow. Emotional symptoms and sleep disturbances should be identified and treated properly since they contribute to the heavy burden experienced by the individual.

information-material/2009/09/public-access-to-information-and-secrecy-act/), but data can be made available for researchers after a special review that includes approval of the research project by both the Swedish Ethical Review Authority (https://etikprovningsmyndigheten.se/en/) and the local authorities' data safety committee in the health care sector in Region Skåne, Sweden; https://vardgivare.skane.se/kompetens-utveckling/forskning-inom-region-skane/utlamnande-av-patientdata-samradkvb/; "Samrådsgrupp för kvalitetsregister, vårddatabaser och beredning"; so called KVB-group) or from the corresponding unit in Region Östergötland, Sweden – according to the "Law for ethical review of research on humans" ["Lag (2003:460) om etikprövning av forskning som avser människor"]. Thus, there are both ethical and legal restrictions on sharing even de-identified data due to potential identification as well as including sensitive information of the interviewed individuals. These restrictions are imposed by the mentioned Swedish Ethical Review Authority (see above). The data set consists of transcribed interviews which are not allowed to be presented publicly as decided in the approval by the Ethical Review Authority.

**Funding:** This work was supported by Grants from Region Östergötland, ALF (Grant number RÖ-977957). The funders had no role in study design, data collection and analysis, decision to publish, or preparation of the manuscript.

**Competing interests:** The authors have declared that no competing interests exist.

## Introduction

The outcome of surgery for ulnar nerve entrapment at the elbow (UNE) is often evaluated using patient-rated outcome measures (PROMs) or objective measures for sensibility and strength [1–3]. However, these may not capture the complexity of UNE in patients and the symptoms and disability that they may experience in conjunction with surgery. Residual or emerging neuropathic pain after surgery for UNE is relatively common [4, 5], but is seldom discussed as an outcome or a complication after surgery in larger studies [6]. Pain affecting the upper limb has additional features compared to pain affecting the lower limb. Functioning of the upper limp is important in human behavior, communication (gestures) and is essential for functional independence in everyday life. Chronic pain in general contributes to a heavy burden, not only for the individual in the form of impaired quality of life [7, 8], but it should also be considered a global public health issue [9] affecting almost 20% of the adult European population [10].

Outcome of surgery for UNE is much more unpredictable compared to surgery for carpal tunnel syndrome for example [11]. A few studies have explored patients' experiences of living with carpal tunnel syndrome and other hand nerve disorders, as well as expectations regarding the results of surgery after carpal tunnel release [12–14]. Our present aim was to explore patients' experiences of living with chronic pain in conjunction with surgery for UNE, its consequences in daily life and the coping strategies used to manage it.

## Method

### Design and participants

A qualitative descriptive method with an inductive approach was used to explore experiences among patients living with chronic pain in conjunction with surgery for UNE. Participants were included from limb tertiary centers, namely; the Hand Surgery, Plastic Surgery and Burns Department at Linköping University Hospital and Hand Surgery at Skåne University Hospital, Malmö, Sweden. Purposive sampling was used to provide a variation regarding sex, age, type of surgery, number of surgeries, educational level/occupational status, and marital status. The inclusion criteria were that participants: had undergone surgery for UNE and had experienced chronic pain (≥3 months duration); were adults (>18 years of age) and could participate in an interview (no serious mental, cognitive and/or linguistic impairments). In total 10 participants were identified and included: five men and five women, with the mean age of 37 years (range 18–60). Different areas of occupation were represented, including e.g., restaurant work, healthcare professions, preschool teaching, industry, firefighting, and customer services. Additional participant characteristics are presented in Table 1. After 10 interviews, no new information was identified from reading the raw data transcripts and the data were considered saturated [15, 16].

### Procedure and ethics

Potentially eligible participants were identified and contacted by telephone by two of the authors (LBD, EN). Prior to the interview, all participants received verbal and written information about the study and provided informed consent, both verbal (documented at the beginning of the interview) and written. Time to consider participating in the study was given.

A semi-structured interview guide (S1 File) was constructed by the authors. All interviews were conducted and tape-recorded by the first author (AG). Participants were recruited and interviewed from February 7th 2022 to May 30th 2022. The first interview was considered a pilot interview, after which all the authors read the transcript to judge whether the interview

**Table 1. Participant characteristics.**

| Participant | Sex | Age | Marital status | Occupation | Affected arm |
|---|---|---|---|---|---|
| 1 | Male | 18–25 | Single | Student | Dominant |
| 2 | Male | 26–40 | Partner | Formerly manual work, now administrative work | Bilateral |
| 3 | Female | 40–50 | Partner | Manual work | Dominant |
| 4 | Female | 50–60 | Single | Early retirement 50%, social assistance 50% | Bilateral |
| 5 | Female | 40–50 | Partner | Manual work, sick leave | Dominant |
| 6 | Female | 18–25 | Partner | Student | Non-dominant |
| 7 | Male | 26–40 | Partner | Manual work, sick leave | Dominant |
| 8 | Female | 40–50 | Partner | Administrative work | Dominant |
| 9 | Male | 40–50 | Partner | Manual work, sick leave | Non-dominant |
| 10 | Male | 18–25 | Partner | Administrative work | Non-dominant |

guide was appropriate and the questions understandable [17]. The pilot interview was included in the data analysis since no major changes were made to the interview guide. The interviews were carried out face-to-face (8 participants) or via Zoom (2 participants) and started with a repetition of the aim of the study. Aside from the two online interviews the interviews were carried out at the Hand Surgery, Plastic Surgery and Burns Department at Linköping University Hospital or at the Department of Hand Surgery at Skåne University Hospital, Malmö, Sweden. The participants were permitted to have another person present but no non-participants were present at the interviews. Mean interview time was 63 minutes, varying from 40 to 87 minutes. The interviews were transcribed verbatim, the first author transcribed the first pilot interview, the remaining interviews were transcribed by one person who was not participating in the study. The transcripts were checked by the first author and any missing words were added.

The study was performed in accordance with relevant guidelines and regulation, including the Declaration of Helsinki, and the National Ethics Review Board (register number 2019–05158) approved the study.

## Data analysis

The text was analyzed using conventional content analysis [18, 19]. The transcripts were read and re-read by the first and second authors (AG and IC) to gain an overview of the content. General impressions were written down as a summary. Meaning units, which are words or sentences related to the aim of the study, were then identified, and coded independently by the same authors. Codes that were similar in content were grouped into subcategories. Subcategories that shared commonalities were identified and abstracted into categories. Examples of the analysis procedure is presented in Table 2. Within each category, similar statements were analyzed critically and questioned, then read and compared until a reasonable interpretation was reached. The categories were then discussed with the third and fourth authors (LBD and EN), who had read all the interviews, and adjustments were made to ensure that the categories covered all aspects of the text (investigator triangulation) [20, 21]. The first author used a

**Table 2. Example of the analysis procedure.**

| Raw data transcript | Meaning unit | Code | Subcategory | Category |
|---|---|---|---|---|
| "If I had to put on socks and a bra for example, I was completely exhausted. Had to catch up. And then, just yes, okay, then we'll take the pants…" | Exhausting to put on socks and bra | Difficulty dressing | Self-care challenges | Consequences in daily life |
| "I drank huge amounts for a period. It was an escape from the pain and to feel a little happy, positive somehow, it was like escaping into another room." | Drinking to escape the pain | Heavy alcohol intake | Pain-relieving strategies | Coping strategies |

qualitative data analysis software to organize transcripts, meaning units, codes, subcategories and categories (Lumivero, Denver, CO (2023) NVivo (Version 12) www.lumivero.com).

Regarding the authors' preunderstanding: the first author was a PhD student in hand surgery, a resident in general medicine and educated in qualitative research; the second author is an experienced occupational therapist specializing in hand rehabilitation, with experience of qualitative research; the third and fourth authors are experienced hand surgeons, familiar with or experienced in qualitative research. None of the participants had previously been treated by the first or second author, but some had been treated by the third or fourth authors.

## Results

The results are presented as seven main categories. *Physical symptoms/impairments* together with *mood and emotions* comprise symptoms caused by UNE and chronic pain. *Consequences in daily life* includes challenges and obstacles in everyday life, impact on leisure activities and social life. *Struggling with self-image* includes experiences closely related to identity. *Coping strategies* covers adaptive resources. *Experience of relief* describes perceived improvements. *Key message for future care* comprises important aspects for healthcare providers to consider. An overview of the main categories and their subcategories is presented in Table 3.

### Physical symptoms/impairments

*Sensory and motor deficits* were described in various ways. Impact on strength, range of motion, grip function and endurance were evident as well as numbness, sensation of "pins and needles" or tingling, spasms/cramps, and loss of sensibility.

**Table 3. Main categories and subcategories.**

| Main categories | Subcategories |
|---|---|
| Physical symptoms/impairments | Sensory and motor deficits |
|  | Pain |
|  | Temperature-related symptoms |
|  | Sleeping disturbances |
|  | Weight changes |
| Mood and emotions | Mood fluctuations and depressive thoughts |
|  | Stress |
|  | Fatigue |
| Consequences in daily life | Self-care challenges |
|  | Difficulties performing household tasks |
|  | Giving up leisure activities |
|  | Feeling isolated and less spontaneous |
| Struggling with self-image | Confronting altered life roles |
|  | Being a burden and experiencing dependency |
| Coping strategies | Pain-relieving strategies |
|  | Activity-related strategies |
|  | Compensation and ergonomics |
|  | Social interaction or isolation |
|  | Asking for help and receiving support |
|  | Emotional and mental processing |
| Experience of relief | - |
| Key message for future care | Information about options and course of treatment |
|  | Being confirmed |

"...this complete numbness was so unpleasant because there was something wrong between the ring finger and the middle finger since these couldn't be felt and the other three sort of existed".

"...weakness in my hands when gripping, screwing, tightening and squeezing is hard"

Claw-hand deformity was also pointed out, as were impacts on fine-motor skills, coordination and problems triggered by vibration.

Different types of *pain*, such as burning, aching, pulsating and sharp pain, were experienced. Allodynia/hyperesthesia also caused unpleasant sensations.

"It was like the nerve kind of rolled somehow. And so, it was like you pulled out a rubber band and then let go. It kind of snapped and it hurt so terribly... The first few times I was absolutely convinced that something had got detached."

When the "pins and needles"/tingling sensations lasted, the pain became more prominent, and it often increased when the participants used their arm/hand.

...before there has been much sharper pain, like knives that cut, like a match that burned me on the forearm. Now it's much more of a meat grinder that is constantly grinding...

Delayed onset of pain after loading the hand was expressed as well as the spreading of pain from a single location to a larger area.

*Temperature-related symptoms*, such as cold and heat sensitivity in the affected arm/hand, were described. These resulted in painful sensations, a sense of coldness, changes in skin colour and sweating. These symptoms fluctuated over time, being more intense some days than others.

"One moment I sweat all over my arm, then it's completely freezing cold, like a piece of ice. And it doesn't matter if it is between 17–18 degrees outside, it can still be freezing cold and it hurts so much right down to the bone."

Current or earlier *sleeping disturbances* with insomnia, frequent nightly awakenings, and a subsequent need to sleep during the day were described.

"I've woken up feeling numb or in a lot of pain and then I've had difficulty getting back to sleep."

These disturbances affected their mood, feelings of fatigue, relationships, and their ability to cope with pain and were reasons for prescriptions of drugs.

The experience of *weight changes* (both gaining and losing weight), due to inability to work out and changes in eating habits, had a great impact on the affected participants. Not being comfortable in their own body led to emotional symptoms, such as depressive thoughts.

"The hardest part was actually after the operation when I couldn't exercise at all. It was absolutely the toughest. I lost 20 kg and yes, it was the worst for my psyche"

## Mood and emotions

A range of emotions were expressed. *Mood fluctuations and depressive thoughts* with frustration, irritability and being short in tone contributed to a bad atmosphere at home and at work.

Not being able to live the life one would like caused frustration and a feeling of sadness. Getting lost in one's own thoughts and feeling indifferent made participants feel as if they had switched off emotionally. A sense of hopelessness, darkness, sadness, and bitterness, as well as lack of energy and a feeling of despair, were also experienced. The feeling of hopelessness even made some question whether life was worth living. Ideas about suicide could occur in the darkest times.

> "Right now I feel pretty bad mentally, pretty far down in some kind of depression. There is no desire, no energy, I have nothing, I am boring, quiet, lost in my own thoughts."

*Stress* was related to worries and uncertainty about the future and to private economic concerns, such as being unable to pay the bills. Not knowing whether they would receive sick pay caused a lot of stress.

> "...they (Swedish Social Insurance Agency) are already starting to tell me to resign and find a new job, but I am on sick leave because I am in pain... and now you have to start otherwise you will be completely excluded from social insurance... and I think, you can't threaten me, I am healing, I have started (working) 25%. It's pretty stressful."

The feeling that the authorities and the healthcare system did not believe them caused stress that led to panic attacks and a feeling of claustrophobia. Not being in charge or able to affect the outcome of surgery together with uncertainty about rehabilitation and symptom progression triggered a great deal of stress.

*Fatigue* could be a major problem, originating in the constant adaptations that had to be made during the day and the increased mental activity needed to avoid risky movements and habitual patterns.

> "Now it's come to the point with the pain that I'm more brain-, not brain-tired maybe...but I'm more exhausted...pain affects me more systematically."

The constant sensation of ongoing pain added to the fatigue. Fatigue made attending lectures difficult for the participants who were students. Avoidance of social events due to exhaustion also occurred. Fatigue was mentioned as a side-effect of prescribed pain medication, which could be a significant problem.

## Consequences in daily life

Showering, dye your hair, putting on jewelry, tying shoes, dressing, and undressing were all parts of *self-care challenges* that the participants struggled with. Buttoning shirts, putting on bras, socks and pants were mentioned as problematic parts of dressing because of fine-motor skill disturbances and fumbling. Lack of energy contributed to difficulties in connection with self-care. Not being able to care for oneself left the participants feeling clumsy and limited compared to others.

> "It is sad. I can't bear to take care of myself anymore."

Various *difficulties when performing household tasks* were also mentioned. Grocery shopping, vacuum cleaning and floor mopping was difficult due to weakness and impaired endurance. Preparing meals was mentioned as problematic because of difficulties in handling cutlery, opening cans, and lifting pots or pans. Impaired sensibility had to be considered when

cutting with a knife. Sometimes the participants had to be cautious and rigorously calculate every single step, to avoid provoking the pain. Others could carry out different tasks without pain but suffered for it later due to delayed pain.

"Try washing. Try to unload the dishwasher. And do it even if it hurts . . . But it won't be good later because it punishes me twice as much later."

Living with pain hindered not only self-care and housework, but also joyful, meaningful activities. Participants had to *give up leisure activities* due to their UNE symptoms, pain, or fatigue. Handiwork, exercising, fixing the car, carpentry and renovating the home, taking care of animals, gardening, calming activities (painting, pearling) and playing guitar were examples mentioned. Even if the symptoms did not stop them from completely abandoning their hobbies, they felt it was a loss not to be able to perform or participate in their usual way. It was described as a great sorrow or boring and that above all else there was not much going on in their life.

"But if you think like, a lot of the other things that I like to do at home or out in the garden, well, it hasn't worked at all. And the same with the stable. I wouldn't have been able to ride or be in the stable. . ."

Turning down invitations from friends or having to give up meaningful activities, meant that participants *felt isolated* or as if they had lost their social context. The fumbling and dropping of items made some feel that they could not dine out due to their poor table manners, which in turn led to cancelling dinner with friends. Another reason for choosing to stay at home was to avoid always having to talk about the arm/hand and explain one's situation. However, valuing social interaction to feel a sense of belonging and experience joy was also emphasized. Having to plan everything according to the daily status of the arm/hand meant that the participants *felt less spontaneous* and thus restricted. Frequently turning down friends and family because of unpredicted mood changes, or sudden onset of fatigue, also contributed to the feeling of being restricted. Reduced spontaneity because of the need to plan activities in advance caused participants to stay at home instead of going to events at short notice, even if they had wanted to.

"It is very boring to plan everything, but I have to, to make it work. . . .Before I would have written in my calendar, yes it's fine. Now, I have to plan to get up earlier like this, take the medicine earlier. Calculate how many hours there are between that medicine and that medicine. What have I planned today? Yes, hmm. Lay out two sets of clothing depending on how the arm feels"

## Struggling with self-image

Living with pain, and the limitations it entails, made the participants question themselves and *confront altered life roles*. Weakness in the arm/hand instilled a sense of insecurity and inability to carry small children. Playing with the children, helping with homework, or escorting them to school could be tiring or impossible. Allowing the children to see their parents vulnerable, or losing their temper in front of them, made the participants question their ability to be a good parent.

"And that I feel that my child suffered, many times. I couldn't go to the swimming pool with her. It was terrible. . . I broke down. I was in complete despair. My child should not have to see this, experience this."

Other life roles were also questioned by the participants. Can one be a good partner despite the unequal distribution of responsibilities and the constantly sad mood? Reduced libido, not wanting to be intimate with one's partner were also mentioned as having a great impact on the participants and their relationships.

"It kind of removes that whole bit, you're like a blank piece of paper. . .it doesn't work as well as it used to so it's tough because I think that in a relationship it's a pretty big part of the whole thing . . . it takes it away, you're simply not as horny anymore." (role as a partner)

Being forced to modify work tasks or change profession completely could be distressing, especially for those in manual work. Too much strain on the arm/hand in such jobs was stressful and forced those participants to move into new fields of work e.g., administrative assignments. The pride in being a competent employee and the feeling of losing one´s work role were then affected. This experience was closely tied to a sense of self and identity in those affected.

"Tough, because that is what you want to keep doing. So, it was like a punch in the stomach, and you have to start over with something else." (work role)

A feeling of insufficiency because of one's limitations and pain affected confidence and self-image. The discrepancy between wanting, but not being able to perform activities, or participate socially due to subsequent pain, limited the participants. Being compelled to ask for help in taking care of basic needs, housework and work tasks instilled a feeling of *being a burden* to those around them. This could even make them feel as if they were not adults because of their *dependency* on others. In accepting adjusted work tasks, they felt they were letting their colleagues down, forcing them to work harder. Being on sick leave and receiving sickness benefit, or having to rely on one's partner financially, also contributed to a feeling of dependency. Such dependency together with not recognizing oneself due to pain limitations gave rise to a struggle with self-image.

"It feels like you're not your own individual who can manage on your own. . . it becomes a bit like you sometimes have a personal assistant in the family. Because you ask for things all the time and you don't really want to do that because it's not really fair."

## Coping strategies

Several *pain-relieving* strategies were mentioned. Distractions in various forms were common, such as watching a movie/TV series, walking, reading a book, mindfulness, relaxation or doing breathing exercises. These distractions enabled them to focus on something other than the pain. While distractions worked for some, others preferred to endure the pain or tried to raise their pain thresholds. Distractions were applied both during the day and at night. When the pain interrupted sleep, some got out of bed, walked back and forth, and swung the arm to relieve the pain before trying to go back to sleep. For others, sleeping was a way of handling the pain, since resting, taking power naps and micropauses made it easier for them to carry out activities during the rest of the day.

Rehabilitative interventions, such as compression gloves, transcutaneous electrical nerve stimulation (TENS), acupuncture and massage, were mentioned as pain-relieving strategies that worked well for some. Finding new ways to exercise or persisting with rehabilitation were

important in keeping the pain, but also one's mental health, under control. Trying to take care of the rest of the body by eating healthy food and exercising could also diminish the pain. Using heat by taking warm baths, using warm blankets, keeping the temperature at home high and sunbathing was a strategy that could alleviate the pain. Choosing clothes that did not provoke allodynia or hyperesthesia was also mentioned.

> "I tried to exercise, I tried to manage my diet, and kind of make sure that I took care of myself physically without straining my left arm, because I've always felt that if I feel good physically, maybe I feel better mentally."

Using alcohol to depress the pain and to avoid boredom and anxiety was mentioned as a strategy that for some had great initial effects. However, the longer the abuse continued the more obvious the consequences of the drinking became, especially when family relationships were affected. A constant sensation of being hung-over, together with increased anxiety, led those participants to question whether this was really a good strategy, and ultimately to a decision to stop drinking.

> "I drank huge amounts for a period. It was an escape from the pain and to feel a little happy, positive somehow, it was like escaping into another room."

Painkillers and sleeping pills of different kinds were prescribed to several of the participants with varying effects. For some it was their salvation, while others had to deal with serious side-effects that outweighed the actual painkilling effect.

*Activity-related strategies*, like pacing or keeping a balance between activity and rest, was a strategy that was adopted. Dividing tasks into smaller steps and taking a break between every interim goal allowed the participants to do more, but at a slower pace. Accepting this slower pace and splitting things up into sub-activities could be frustrating.

> "If I'm going to clean at home, I really have to have a whole cleaning day because I can't do everything at once, then I end up in so much pain that the whole arm goes numb from pain. But I can sort of divide it up, do something now and something a little later"

Planning became important because of the change in the occupational balance. Some had to take every possible mood or outcome into account before engaging in certain tasks. Considering whether some action would trigger the pain before doing it, made the participants aware of possible risk activities. Driving to work outside rush hours to avoid heavy traffic or dictating instead of using a computer keyboard to avoid extensive load on the arm/hand were strategies used to change activity patterns or performances. Continued participation and routines despite pain and possible restrictions, especially when on sick leave, enabled the participants to stay involved.

*Compensations and ergonomic* adaptations were also employed. Using the non-injured arm/hand or both hands was a common strategy for dealing with both pain and sensory and motor symptoms. Some even mentioned that using this strategy improved the dexterity in their non-dominant hand. This strategy, however, could have an opposite effect in the long run by overloading the non-injured arm. Using assistive devices, such as can openers or drinking straws, instead of lifting a glass full of water, and choosing simpler clothing, were other compensatory adaptations. Holding heavy objects close to the body and keeping the elbow stable were ergonomic measures that were mentioned. Listening to audiobooks or reading books on an electronic device was a way in which to consume fiction without having to hold a heavy

book. Pillows were used to create a comfortable position in which the participant could fall asleep or rest the arm when driving. Placing objects at a comfortable height, e.g. in kitchen cabinets, enabled the participants to grasp them.

> "..keep things close to your body so that you get stability from it. . .and to have things placed in the right direction so you can just take it and not have to twist it. . .

A conscious choice of *social interaction or isolation* was described. Keeping in touch with friends and family and keeping up a social network made some feel less alone and less of an outsider. It could also serve as a distraction from the pain and prevent distress. Others limited and narrowed their social network, only retaining the most meaningful relationships, to avoid being questioned and judged by distant acquaintances. Ordering food online was one way to avoid meeting acquaintances at the store.

> "I isolated myself. There were almost depressive tendencies in my behavior that I did not go out. I did not socialize with people in my vicinity either and barely shopped"

A common strategy for overcoming challenges in daily life was *asking for help* from family, friends, or colleagues. Some did not need to ask for help, as their relatives or friends noticed their struggles and came to assist.

> ". . .she became almost overprotective of me, you don't have to clean, don't do it, I'll fix it. Then I felt like I was nobody, that I couldn't do anything. . ..You kind of become an arm, I myself disappeared. . ."

Others wanted more help from family, especially immediately after the surgery. This *support* took many forms, ranging from getting help with basic needs, such as dressing, preparing food, and taking care of the household, to support that were more emotional. Receiving verbal support from family and friends was not only considered to be extremely valuable and providing relief, but also made some participants feel that they were always nagging, pessimistic and casting a gloomy shadow over the other person's day. Explaining daily physical or emotional mood could be tiring.

> "I have explained pain and such and we have tried to find strategies like this—yes it is a yellow day today, or a red day today."

Others did not want to be questioned about their arm/hand at all and only brought it up themselves.

Seeking help and support from healthcare was another strategy for handling both uncertainty regarding the outcome of the surgery or rehabilitation, and emotional symptoms. Some went to therapy to sort out their thoughts and learn useful coping strategies, such as realizing that this condition does not define you and that life is worth living, despite possible restrictions. Dealing with other emotional issues that did not directly correlate to pain, such as those related to family, was also helpful and made it easier to tackle the symptoms. Repeatedly calling hand surgical clinics or pain clinics, and getting a timeline for the next procedure, were other ways of accessing support. Online communities were another forum that provided support. Talking online to people with similar experiences and getting feedback from others made the participants realize that they were not alone.

Dealing with challenges in everyday life required *emotional and mental processing*. For the participants this meant realizing and accepting their situation, despite possible limitations, or holding on to the hope that everything would be alright in the end. Some hoped further surgery would fix the whole situation, while others pinned their hopes on rehabilitation.

"I don't feel ready, but my doctor is starting to feel ready with me. But I'm not done. Because I don't want to live with this pain."

Positive thinking worked to a certain degree, but after a while a feeling of indifference could set in. Focusing on progress over time helped some participants to stay positive, and to gain insight into their own mental growth. Trying to push the limits, striving, but in moderation, helped the participants to keep on fighting through the hardest of times. Therapy helped them to realize that physical and mental health are closely connected and affect one another. Writing a diary or posting on social media were ways of reflecting and being able to let go of the past. Comparing your situation with something even worse and thus minimizing your own problems, focusing on what you have and what works, were other strategies mentioned. Seeking information and intellectualizing experiences were conscious actions towards comprehending the situation. A mental process about what is important and worth focusing on also helped them to move forward and let go.

"I did a lot of mental processing—came out of the rain clouds and it was like a little bit above the clouds and it was kind of sunny somehow—mentally, so a lot fell into place. . .and I realized that I will not be 100% recovered and had to process it for a while".

### Experience of relief

A strong sense of *relief* was expressed by those participants who experienced improvements after surgery, although some still suffered with chronic pain and its consequences. Regaining strength, range of motion and sensibility as well as reduced paresthesia and pain were mentioned as results of surgery.

"If I had 100% strength before I had the surgery, maybe I was down to 40% after the surgery was done, then maybe I'm up to 60–70% now, so I still have a bit to go until I feel like I'm as strong and can do what I did before. Can still say—a very good curve. And I would say that it is pretty much as it should be, when you are born a human."

Others experienced physical or emotional relief through rehabilitative interventions, conversational or cognitive behavioral therapy or through the use of other coping strategies. Noticing improvements boosted hope and confidence.

"Being able to put on a bra and stockings or button a pair of pants, just like that, a rush of happiness and that I didn't constantly wake up 10 times during the night, because my arm was asleep."

Improved sleep was crucial and enabled participants gradually, and to a varying degree, to recapture the ability to perform daily activities. Coming to terms with the current situation, accepting or managing the pain helped the participants to move forward and focus on what could be improved or changed. This change of mindset also contributed to a sense of relief.

### Key message for future care

Additional *information about options and courses of treatment* were requested by some participants. The importance of being able to make an informed decision about whether to undergo surgery was emphasized. Participants wanted to receive individualized, comprehensible information regarding the condition and its pathophysiology. They also wanted realistic outcome statistics and clear information about possible complications after surgery. If they had known about all the possible complications or the potential worsening of their symptoms, some would have rejected surgery and chosen other treatment options.

> "That's what I would like to say—think again if you really want to (undergo surgery). No matter how big the troubles, can you handle them now? Try something else first as well. Therapy, talking, try other things first before an operation."

Others stated the opposite, that despite the outcome being that the situation was exacerbated, they would have gone through the same procedures again. Some felt that they had been treated as guinea pigs as they felt the surgeon did not know how to treat them. Some did not want to rush to have surgery or pain medication, but instead wanted to discuss other ways of treating recurring pain, for example, as was suggested, by talking to someone about how one could deal with the pain. One potential improvement mentioned was that information should be given about the possible side-effects of medication, especially regarding reduced libido.

> "Yes, of course, you can read the package leaflets, absolutely, but yes, you still want to have it explained to yourself a little bit, just so you know how this part (libido) can be affected, and it's a big part, so. . ."

To *be confirmed* by the healthcare professionals was said to be an important aspect. The invisibility of pain as a symptom and the absence of obvious signs of pain or fatigue was frustrating. Being believed and confirmed was therefore crucial.

> "Then it feels so great to have a doctor who understands. Well, I understand that you have this even if I can't feel what you feel. I believe in you. And to know that he believes in me and to know that my employer believes in me and to know that my family understands and sees what it's like, and then. . . it does a lot. So, that is how I deal with it."

> "When you get it confirmed. . ., it's like the air goes out of you. You just lower your shoulders and just, oh. Like you have held your breath for 10 years, so don't give up! There is, there. . .she has enchanted my arm."

In contrast, not being believed by the healthcare professionals made some wonder if they were hypochondriacs. They wanted to be seen as more than an arm/hand that cannot be fixed, but rather as a human being with feelings.

## Discussion

This qualitative study explores patients' experiences of living with chronic pain in conjunction with surgery for UNE and reveals a many-sided description of the physical symptoms, mood and emotions, consequences in daily life and adaptations used by the participants. Living with chronic pain affects self-image, through changes in life roles, and gives rise to a feeling of being a burden. Coping strategies of various kinds were employed by the participants and relief was

sometimes experienced. The participants communicated aspects that healthcare providers need to consider.

The participants described not only typical physical symptoms of UNE, but also emotional symptoms. Depressive thoughts and poor mental health are not uncommon among patients with chronic pain and the relationship can be seen as bidirectional, both resulting from and causing one another [22]. The combination of depression and chronic pain can be described as a vicious cycle, where depression increases pain leading to reduced physical activity, which in turn increases pain [7]. The importance of continued exercise, and taking care of the body in general, were both mentioned as important coping strategies. Not being able to exercise as usual had a great impact on the participants' mental health and was described as a difficult aspect to deal with.

In a previous study, using data from questionnaires, we found that patients living with chronic pain after surgery for UNE have low life satisfaction, a high degree of psychological distress and low overall health status [4]. Psychological intervention is an important part of multidisciplinary care and can function as an alternative to medication, depending on the severity and nature of the pain [23]. Multimodal rehabilitation programs (MMRP) of different designs are based on a biopsychosocial model of chronic pain [24], and usually involve patient education, supervised physical activity, cognitive behavioral therapy, and work-related efforts. The components are coordinated and delivered by a team of professionals with the patient as an active participator [24, 25]. Real-life studies of MMRP show moderate and stable pain reduction and better state of health at 12-month follow-up [26]. This more holistic or biopsychosocial perspective was desired by the present participants, who felt that their mental health was being neglected. Referrals for MMRP could therefore be considered.

Other prominent symptoms among the present participants were sleeping disturbances and fatigue due to pain. The prevalence of sleeping disturbances among patients with chronic pain is generally high [27]. Sleeping disturbances have been associated with greater pain severity, longer pain duration and greater disability in patients with chronic pain [28]. Patients with both sleeping disturbances and chronic pain are more likely to have concurrent depression, anxiety, and suicidal ideations [28]. Anxiety concerns over money and lack of control over the outcome or the future were experienced by our participants, which probably added to their sleeping disturbances. Suicidal ideations and reduced joy of life were also reported, emphasizing the great impact this condition can have. Being able to sleep was mentioned as the turning point by some, who explained that they got their life back when they could sleep. The sleep perspective has also been emphasized in a previous qualitative study of patients following carpal tunnel release (27% of patients) or particularly ulnar nerve decompression at the elbow (73% of patients) 6–24 months earlier. A good night's sleep due to relief of nocturnal numbness was expressed in those patients as the most satisfactory improvement, despite that, in contrast to our by pain severely affected subjects, their patients did not perceived their conditions as serious [29]. In one of our previous studies, we found that 68% of patients with UNE had preoperative sleeping problems, which improved postoperatively for 58% [5]. Preoperative sleeping disturbances have been shown to be a significant predictor of chronic post-surgical pain [30]. Addressing possible sleeping disturbances is, therefore, of great importance in clinical practice as it contributes to morbidity.

Struggling with self-image formed a central part of the narratives. Having to rely on others in activities that one had previously managed independently induces a feeling of insufficiency. Due to this, it was common for the participants to question themselves in their role of partner, parent, and colleague. The shift in life-roles, the feeling of being a burden as well as fear of independence have been described in other hand-related disorders as well as in patients with chronic pain [13, 31–33]. Having to move into a new field of work can be challenging, since

work and identity are closely connected [34]. This was experienced by the present participants who felt that their self-image was affected when they had to find alternative work tasks or jobs. Employment also helps to structure the day, which became evident in participants on sick leave who mentioned the need to keep routines when they were not working. Considering the patients' different life roles or levels of dependency on others is important when supporting patients' adaptation. Embracing a "top-down" approach, meaning that the patients' roles, habits, and activities are used as a starting point for care, may facilitate a more biopsychosocial approach [35].

The participants in our study used several different coping mechanisms. There are no universally good or universally bad coping strategies; everything depends on the context in which they are used. Coping, defined as cognitive and behavioral efforts to manage psychological stress, can be divided into problem-based coping and emotion-focused coping. Problem-based coping focuses on altering the situation that causes the distress and is more commonly used when the situation is likely to change. Emotion-based coping focuses on regulating the emotions caused by the distress and is more commonly used when a situation is unlikely to change [36, 37]. The present participants used both problem- and emotion-based coping. They described using the non-affected arm/hand as a compensatory strategy with a subsequent overload on that side, this is also illustrated in women with CTS [12]. Other examples of coping strategies used by our participants were distractions, pacing and asking for help. These strategies are some of the most used to manage chronic pain [38]. Changing activity pattern or performance have also been mentioned as coping strategies when dealing with other hand-related issues [39]. Different types of coping mechanisms have been linked to varying degrees of distress (measured by the Hospital Anxiety and Depression Scale, HADS) among patients with an acute traumatic hand injury. Patients without emotional distress tend to accept the situation and minimize the problem more often than patients with emotional distress, who used more confrontational and emotive coping strategies [40]. Emotional distress was expressed in different ways by our participants. Confrontational and emotive coping strategies, such as relieving emotional tension through physical activity, worrying, getting nervous or angry, were recurrently mentioned by the participants in our study.

The invisibility of pain due to UNE was described as frustrating and could contribute to a feeling of not being believed. Women with CTS have described using wristbands as a way of combating disbelief among others in their environment [12]. The feeling of not being believed by healthcare professionals, due to the inability to measure pain objectively, is not unique to hand-related pain but occur across different chronic pain conditions [32]. In contrast, our participants described being confirmed by healthcare professionals as an important aspect that brought alleviation. Validating the experience, making patients feel understood and heard is therefore something that a hand surgeon should incorporate into daily practice [41]. Shared decision making can improve patients' knowledge about treatment options and satisfaction, especially when they feel involved [42, 43]. Improving the information about postoperative care and expectations of recovery was also considered important for surgically treated patients with carpal tunnel syndrome or mainly with ulnar nerve entrapment at elbow, as described in a previous qualitative study [29].

Two key factors may be important for maintaining health when dealing with stressors—the generalized resistance resources and the Sense of Coherence (SOC). The resistance resources reinforce the SOC and help us to comprehend, manage and find meaning in the movement towards positive health [44]. Our participants experienced both ability and inability to understand their given situation or symptoms. The unpredictable nature of pain also seemed illogical and overwhelming to some (comprehensibility). Experiencing unpredictability is a common phenomenon across different pain conditions [32]. Realizing that life will not be the same as

before, and then trying to make the best of the new situation, was described by those participants who had access to effective resistance resources, e.g., coping strategies. Despite the rich array of coping strategies presented in the narratives, the participants still suffered from chronic pain and experienced an imbalance between demands and resources (manageability). The feeling of being a burden and having to give up joyful activities because of UNE symptoms, pain or fatigue were commonly described (meaningfulness). The lack of comprehensibility, manageability and meaningfulness contributes to a lower SOC and thereby a lower perceived level of health. Studies of major hand injuries show that patients with low SOC have significantly lower satisfaction with daily occupations, lower mental quality of life, more sleep disturbances and bodily pain [45]. Taking SOC into account to identify patients who need extra support may therefore be of importance in clinical practice.

## Methodological considerations

In qualitative research, the findings are evaluated in terms of trustworthiness, which includes establishing credibility, dependability, confirmability, and transferability [46, 47]. A detailed description of methods used to strengthen trustworthiness is presented in Table 4. The participants have varying life experiences, comorbidity, personality traits and backgrounds.

Recall bias can always be a concern when discussing previous experiences, however most of the participants described strong memories of past experiences and many still lived with pain. Even if some details may have been altered by recall bias it is not likely to have influenced the overall findings. Depressive thoughts were commonly mentioned and some of the participants may even have been clinically depressed, this may have influenced their answers regarding experiences and coping strategies. However, these experiences are valuable since many of the participants go through a challenging time mentally when dealing with chronic pain. This is a perspective that we wanted to highlight and therefore only excluded participants with considerable mental impairments. Our intention in the present study was to describe the experiences

**Table 4. Methods used in the present study to evaluate trustworthiness.**

| Trustworthiness | |
|---|---|
| Credibility | Purposive sampling was used to reflect and broaden different experiences of living with chronic pain in conjunction with UNE. The number of participants was limited; however, the interviews were all rich in detail. The method used consistently was an analysis that focused on the content of the text and related to the aim, limiting the risk of predetermined interpretation. |
| | The categories reflect the transcribed text and the aim of the study. Similarities in content were grouped into categories and clear differences between categories were emphasized to reduce risk of overlap. |
| | Member check [47, 48]: Credibility was ensured by constantly confirming and clarifying information during the interviews. |
| Dependability and confirmability | Investigator triangulation [20, 21, 48]: To strengthen dependability and reduce the risk of over interpretation of the results due to the authors' pre-understanding of the phenomena in focus, two authors independently read and coded the text and engaged in in-depth discussion to arrive at a reasonable interpretation. Furthermore, the third and last authors read all interviews and confirmed the interpretation and categorization. |
| | Representative quotations from the transcribed text are presented to make the interpretation of the text visible to the reader. |
| | The authors preunderstandings are stated in the methods section. |
| Transferability | A detailed ("thick") description [47, 49] of the study group was presented in the methods section. The participants represent a small study group, which limits the transferability of the findings. |

of living with chronic pain in conjunction with surgery for UNE irrespective of whether it occurred before or after the surgery, or in some cases between surgical sessions. Categories are therefore not divided into pre- or post-operative experience.

## Conclusion

Patients with chronic pain in conjunction with surgery for UNE experience a wide variety of symptoms. Pain, sleeping disturbances and emotional symptoms have a great impact on daily life and should be recognized and managed early on since they contribute to the heavy burden for the individual. Access to various coping strategies can to a certain degree alleviate symptoms, reduce limitations in daily life and be of importance when struggling with self-image. Key message for future care includes the need for patients to be confirmed and receive individualized information about treatment options. A biopsychosocial perspective among healthcare personnel, can contribute to a more holistic approach where chronic pain and its effect on all aspects of daily life are taken into consideration.

## Supporting information

**S1 File. Semi-structured interview guide.**
(PDF)

## Author Contributions

**Conceptualization:** Alice Giöstad, Ingela K. Carlsson, Lars B. Dahlin, Erika Nyman.

**Data curation:** Alice Giöstad, Erika Nyman.

**Formal analysis:** Alice Giöstad, Ingela K. Carlsson.

**Funding acquisition:** Alice Giöstad, Erika Nyman.

**Investigation:** Alice Giöstad, Ingela K. Carlsson.

**Methodology:** Alice Giöstad, Ingela K. Carlsson, Lars B. Dahlin.

**Project administration:** Lars B. Dahlin, Erika Nyman.

**Resources:** Lars B. Dahlin, Erika Nyman.

**Software:** Alice Giöstad.

**Supervision:** Ingela K. Carlsson, Lars B. Dahlin, Erika Nyman.

**Validation:** Ingela K. Carlsson, Lars B. Dahlin.

**Visualization:** Alice Giöstad, Ingela K. Carlsson.

**Writing – original draft:** Alice Giöstad.

**Writing – review & editing:** Alice Giöstad, Ingela K. Carlsson, Lars B. Dahlin, Erika Nyman.

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
