## [Decision Letter · Decision Letter 0]

8 Mar 2024

PONE-D-23-35335Experience of living with chronic pain in conjunction with surgery for ulnar nerve entrapment at the elbow – a qualitative studyPLOS ONE

Dear Dr.Alice Giöstad,

Thank you for submitting your manuscript to PLOS ONE. After careful consideration, we feel that it has merit but does not fully meet PLOS ONE’s publication criteria as it currently stands. Therefore, we invite you to submit a revised version of the manuscript that addresses the points raised during the review process.

We look forward to receiving your revised manuscript.

Kind regards,

Priti Chaudhary, M.S.

Academic Editor

PLOS ONE

Journal Requirements:

Reviewers' comments:

Reviewer's Responses to Questions

**Comments to the Author**

1. Is the manuscript technically sound, and do the data support the conclusions?

Reviewer #1: Yes

Reviewer #2: Partly

2. Has the statistical analysis been performed appropriately and rigorously? 

Reviewer #1: Yes

Reviewer #2: N/A

3. Have the authors made all data underlying the findings in their manuscript fully available?

Reviewer #1: No

Reviewer #2: Yes

4. Is the manuscript presented in an intelligible fashion and written in standard English?

Reviewer #1: Yes

Reviewer #2: Yes

5. Review Comments to the Author

Reviewer #1: Overall, it is an interesting work discuss the management challenges for an uncommon condition.

Conclusion can be made more informative and needs to be further expanded to reflect the findings from the current study.

Reviewer #2: Review report

Dear authors

I have thoroughly reviewed your article on addressing a qualitative study that after ulnar nerve surgery. What are the differences between your study and '' Kathleen Joy Khu at all. Patients’ Perceptions of Carpal Tunnel and Ulnar Nerve Decompression Surgery ''?

https://www.cambridge.org/core/journals/canadian-journal-of-neurological-sciences/article/patients-perceptions-of-carpal-tunnel-and-ulnar-nerve-decompression-surgery/FFF7A6C677FB8354C92E520437998615

Best regards

6. PLOS authors have the option to publish the peer review history of their article (what does this mean?). If published, this will include your full peer review and any attached files.

Reviewer #1: No

Reviewer #2: No

---

## [Author Response · Author response to Decision Letter 0]

2 Apr 2024

Question: Is the manuscript technically sound, and do the data support the conclusions?

Reviewer #1: Yes

Reviewer #2: Partly

Reply: No comment required.

Question: Has the statistical analysis been performed appropriately and rigorously? 

Reviewer #1: Yes

Reviewer #2: N/A

Reply: No comment required.

Question: Have the authors made all data underlying the findings in their manuscript fully available?

Reviewer #1: No

Reviewer #2: Yes

Reply: We have included a separate comment concerning this issue based on the approval by our Ethical committee:

“Data availability statement

Public access to research data on humans is restricted by the Swedish Authorities (Public Access to Information and Secrecy Act; https://www.government.se/information-material/2009/09/public-access-to-information-and-secrecy-act/), but data can be made available for researchers after a special review that includes approval of the research project by both the Swedish Ethical Review Authority (https://etikprovningsmyndigheten.se/en/) and the local authorities' data safety committee in the health care sector in Region Skåne, Sweden; https://vardgivare.skane.se/kompetens-utveckling/forskning-inom-region-skane/utlamnande-av-patientdata-samradkvb/; “Samrådsgrupp för kvalitetsregister, vårddatabaser och beredning”; so called KVB-group) or from the corresponding unit in Region Östergötland, Sweden – according to the “Law for ethical review of research on humans” [“Lag (2003:460) om etikprövning av forskning som avser människor”]. Thus, there are both ethical and legal restrictions on sharing even de-identified data due to potential identification as well as including sensitive information of the interviewed individuals. These restrictions are imposed by the mentioned Swedish Ethical Review Authority (see above). The data set consists of transcribed interviews which are not allowed to be presented publicly as decided in the approval by the Ethical Review Authority. “

Question: Is the manuscript presented in an intelligible fashion and written in standard English?

Reviewer #1: Yes

Reviewer #2: Yes

Reply: Thank you.

Reviewer #1: 

Question: Overall, it is an interesting work discuss the management challenges for an uncommon condition.

Conclusion can be made more informative and needs to be further expanded to reflect the findings from the current study.

Reply: Changes have been made in conclusion.

Reviewer #2: Review report

Question: I have thoroughly reviewed your article on addressing a qualitative study that after ulnar nerve surgery. What are the differences between your study and '' Kathleen Joy Khu at all. Patients’ Perceptions of Carpal Tunnel and Ulnar Nerve Decompression Surgery ''?

https://www.cambridge.org/core/journals/canadian-journal-of-neurological-sciences/article/patients-perceptions-of-carpal-tunnel-and-ulnar-nerve-decompression-surgery/FFF7A6C677FB8354C92E520437998615

Reply: Thank you for highlighting this study, where patients with carpal tunnel syndrome or mainly ulnar nerve entrapment at the elbow were interviewed 6-24 months after surgery. We have added the reference in our Discussions at two locations and made a slight comparison; however, their study differ substantially to ours. They emphasize based on their interviews that “(1) most patients did not perceive their condition to be serious; (2) patients were satisfied with the overall surgical experience; (3) the outcome was more important to patients than the process (4) majority of patients had a realistic expectation of outcomes.” Furthermore, they concluded that their patients “had a positive experience with carpal tunnel and ulnar nerve decompression surgery, although their level of satisfaction was dependent on the surgical outcome.”. Our patients had solely undergone surgery of the ulnar nerve entrapment at the elbow and our focus was on pain problems, which makes it difficult to make a thorough comparison. Therefore, we have only made a limited comparison. 

We have included an updated statement for financial disclosure in the Cover letter. Updated statement for financial disclosure: This work was supported by Grants from Region Östergötland, ALF (Grant number RÖ-977957). The funders had no role in study design, data collection and analysis, decision to publish, or preparation of the manuscript.

Sincerely yours, 

Alice Giöstad on behalft of the co-authors

---

## [Editor Report · Decision Letter 1]

12 Apr 2024

PONE-D-23-35335R1Experience of living with chronic pain in conjunction with surgery for ulnar nerve entrapment at the elbow – a qualitative studyPLOS ONE

Dear Dr. Giöstad,

Thank you for submitting your manuscript to PLOS ONE. After careful consideration, we feel that it has merit but does not fully meet PLOS ONE’s publication criteria as it currently stands. Therefore, we invite you to submit a revised version of the manuscript that addresses the points raised during the review process.

**Authors are required to reply:**

**1. Sample size calculation not done. Kindly explain.**

**2. Sample size is very small (10). Kindly justify.**

**3. Nothing is mentioned about validation of semi-structured interview guide. Kindly explain.**

We look forward to receiving your revised manuscript.

Kind regards,

Priti Chaudhary, M.S.

Academic Editor

PLOS ONE
---

## [Author Response · Author response to Decision Letter 1]

16 Apr 2024

Questions:

1. Sample size calculation not done. Kindly explain.

Reply: Sample size calculation is common in quantitative research and sample size is most often determined before the start of the data collection for quantitative research questions. However, qualitative research studies uses smaller sample sizes but provides information in greater depth. Instead of calculating sample size beforehand, the sampling continues until data saturation has been achieved. Data saturation is reached when no new analytical information arises anymore. Two references is added in the methods section to clarify this approach. 

2. Sample size is very small (10). Kindly justify.

Reply: The sampling in this study was made deliberately with purposive sampling which is a common method in qualitative research. In purposive sampling the researchers selects the participants that will be most informative and provide different perspectives of the studied phenomenon. We reached data saturation after ten interviews which was possible due to the information richness in the data and the variety of the participants. 

3. Nothing is mentioned about validation of semi-structured interview guide. Kindly explain.

Reply: The semi-structured interview guide was developed in several steps and with care by the authors. The format of a semi-structured interview was considered as an appropriate method for data collection for this research question. By retrieving and using previous knowledge we gained a comprehensive and adequate understanding of the phenomenon under study. After this we formulated the preliminary semi-structured interview guide that was field tested in a pilot-interview with a potential study participant. The pilot interview was analysed to ensure that the research questions were being covered and that the questions were understandable. This approach is common in qualitative research. A reference have been added to the methods section to clarify this approach. 

We do hope that these changes are sufficient and that you now can consider the manuscript suitable for publication.

Sincerely yours

Alice Giöstad Ingela K. Carlsson Lars B. Dahlin Erika Nyman

---

## [Decision Letter · Decision Letter 2]

17 Jun 2024

Experience of living with chronic pain in conjunction with surgery for ulnar nerve entrapment at the elbow – a qualitative study

PONE-D-23-35335R2

Dear Dr. Alice Giöstad,

We’re pleased to inform you that your manuscript has been judged scientifically suitable for publication and will be formally accepted for publication once it meets all outstanding technical requirements.

Kind regards,

Priti Chaudhary, M.S.

Academic Editor

PLOS ONE

Additional Editor Comments (optional):

Reviewers' comments:

Reviewer's Responses to Questions

**Comments to the Author**

1. If the authors have adequately addressed your comments raised in a previous round of review and you feel that this manuscript is now acceptable for publication, you may indicate that here to bypass the “Comments to the Author” section, enter your conflict of interest statement in the “Confidential to Editor” section, and submit your "Accept" recommendation.

Reviewer #1: All comments have been addressed

2. Is the manuscript technically sound, and do the data support the conclusions?

Reviewer #1: Yes

3. Has the statistical analysis been performed appropriately and rigorously? 

Reviewer #1: Yes

4. Have the authors made all data underlying the findings in their manuscript fully available?

Reviewer #1: Yes

5. Is the manuscript presented in an intelligible fashion and written in standard English?

Reviewer #1: Yes

6. Review Comments to the Author

Reviewer #1: Authors have made suggested changes.

7. PLOS authors have the option to publish the peer review history of their article (what does this mean?). If published, this will include your full peer review and any attached files.

Reviewer #1: No

---

## [Editor Report · Acceptance letter]

20 Jun 2024

PONE-D-23-35335R2 

PLOS ONE

Dear Dr. Giöstad, 

I'm pleased to inform you that your manuscript has been deemed suitable for publication in PLOS ONE. Congratulations! Your manuscript is now being handed over to our production team.

Kind regards, 

on behalf of

Dr. Priti Chaudhary 

Academic Editor

PLOS ONE